# Private Graphon Estimation for Sparse Graphs[*]

**Christian Borgs**     **Jennifer T. Chayes**
Microsoft Research New England
Cambridge, MA, USA.
`{cborgs,jchayes}@microsoft.com`

**Adam Smith**
Pennsylvania State University
University Park, PA, USA.
`asmith@psu.edu`

## Abstract

We design algorithms for fitting a high-dimensional statistical model to a large, sparse network without revealing sensitive information of individual members. Given a sparse input graph $G$, our algorithms output a node-differentially private nonparametric block model approximation. By node-differentially private, we mean that our output hides the insertion or removal of a vertex and all its adjacent edges. If $G$ is an instance of the network obtained from a generative nonparametric model defined in terms of a graphon $W$, our model guarantees consistency: as the number of vertices tends to infinity, the output of our algorithm converges to $W$ in an appropriate version of the $L_2$ norm. In particular, this means we can estimate the sizes of all multi-way cuts in $G$.

Our results hold as long as $W$ is bounded, the average degree of $G$ grows at least like the log of the number of vertices, and the number of blocks goes to infinity at an appropriate rate. We give explicit error bounds in terms of the parameters of the model; in several settings, our bounds improve on or match known nonprivate results.

## 1   Introduction

**Differential Privacy.** Social and communication networks have been the subject of intense study over the last few years. However, while these networks comprise a rich source of information for science, they also contain highly sensitive private information. What kinds of information can we release about these networks while preserving the privacy of their users? Simple measures, such as removing obvious identifiers, do not work; for example, several studies reidentified individuals in the graph of a social network even after all vertex and edge attributes were removed. Such attacks highlight the need for statistical and learning algorithms that provide rigorous privacy guarantees.

Differential privacy [17] provides meaningful guarantees in the presence of arbitrary side information. In the context of traditional statistical data sets, differential privacy is now well-developed. By contrast, differential privacy in the context of graph data is much less developed. There are two main variants of graph differential privacy: *edge* and *node* differential privacy. Intuitively, edge differential privacy ensures that an algorithm's output does not reveal the inclusion or removal of a particular edge in the graph, while node differential privacy hides the inclusion or removal of a node together with all its adjacent edges. Edge privacy is a weaker notion (hence easier to achieve) and has been studied more extensively. Several authors designed edge-differentially private algorithms for fitting generative graph models (e.g. [24]; see the full version for further references), but these do not appear to generalize to node privacy with meaningful accuracy guarantees.

The stronger notion, node privacy, corresponds more closely to what was achieved in the case of traditional data sets, and to what one would want to protect an individual's data: it ensures that *no matter what an analyst observing the released information knows ahead of time*, she learns the same

---

[*]A full version of this extended abstract is available at `http://arxiv.org/abs/1506.06162`

things about an individual Alice regardless of whether Alice's data are used or not. In particular, no assumptions are needed on the way the individuals' data are generated (they need not even be independent). Node privacy was studied more recently [21, 14, 6, 26], with a focus on on the release of descriptive statistics (such as the number of triangles in a graph). Unfortunately, differential privacy's stringency makes the design of accurate, private algorithms challenging.

In this work, we provide the first algorithms for node-private inference of a high-dimensional statistical model that does not admit simple sufficient statistics.

**Modeling Large Graphs via Graphons.** Traditionally, large graphs have been modeled using various parametric models, one of the most popular being the stochastic block model [20]. Here one postulates that an observed graph was generated by first assigning vertices at random to one of $k$ groups, and then connecting two vertices with a probability that depends on their assigned groups.

As the number of vertices of the graph in question grows, we do not expect the graph to be well described by a stochastic block model with a fixed number of blocks. In this paper we consider nonparametric models (where the number of parameters need not be fixed or even finite) given in terms of a *graphon*. A graphon is a measurable, bounded function $W : [0, 1]^2 \to [0, \infty)$ such that $W(x, y) = W(y, x)$, which for convenience we take to be normalized: $\int W = 1$. Given a graphon, we generate a graph on $n$ vertices by first assigning i.i.d. uniform labels in $[0, 1]$ to the vertices, and then connecting vertices with labels $x, y$ with probability $\rho_n W(x, y)$, where $\rho_n$ is a parameter determining the density of the generated graph $G_n$ with $\rho_n \|W\|_\infty \le 1$. We call $G_n$ a $W$-random graph with target density $\rho_n$ (or simply a $\rho_n W$-random graph).

To our knowledge, random graph models of the above form were first introduced under the name latent position graphs [19], and are special cases of a more general model of "inhomogeneous random graphs" defined in [7], which is the first place were $n$-dependent target densities $\rho_n$ were considered. For both dense graphs (whose target density does not depend on the number of vertices) and sparse graphs (those for which $\rho_n \to 0$ as $n \to \infty$), this model is related to the theory of convergent graph sequences, [8, 23, 9, 10] and [11, 12], respectively.

**Estimation and Identifiability.** Assuming that $G_n$ is generated in this way, we are then faced with the task of estimating $W$ from a *single observation* of a graph $G_n$. To our knowledge, this task was first explicitly considered in [4], which considered graphons describing stochastic block models with a fixed number of blocks. This was generalized to models with a growing number of blocks [27, 15], while the first estimation of the nonparametric model was proposed in [5]. Most of the literature on estimating the nonparametric model makes additional assumptions on the function $W$, the most common one being that after a measure-preserving transformation, the integral of $W$ over one variable is a strictly monotone function of the other, corresponding to an asymptotically strictly monotone degree distribution of $G_n$. (This assumption is quite restrictive: in particular, such results do not apply to graphons that represent block models.) For our purposes, the most relevant works are Wolfe and Olhede [28], Gao et al. [18], Chatterjee [13] and Abbe and Sandon [2] (as well as recent work done concurrently with this research [22]), which provide consistent estimators without monotonicity assumptions (see "Comparison to nonprivate bounds", below).

One issue that makes estimation of graphons challenging is *identifiability*: multiple graphons can lead to the same distribution on $G_n$. Specifically, two graphons $W$ and $\tilde{W}$ lead to the same distribution on $W$-random graphs if and only if there are measure preserving maps $\phi, \tilde{\phi} : [0, 1] \to [0, 1]$ such that $W^\phi = \widetilde{W}^{\tilde\phi}$, where $W^\phi$ is defined by $W(x, y) = W(\phi(x), \phi(y))$ [16]. Hence, there is no "canonical graphon" that an estimation procedure can output, but rather an equivalence class of graphons. Some of the literature circumvents identifiability by making strong additional assumptions, such as strict monotonicity, that imply the existence of canonical equivalent class representatives. We make no such assumptions, but instead define consistency in terms of a metric on these equivalence classes, rather than on graphons as functions. We use a variant of the $L_2$ metric,

$$\delta_2(W, W') = \inf_{\phi:[0,1]\to[0,1]} \|W^\phi - W'\|_2, \text{ where } \phi \text{ ranges over measure-preserving bijections.} \quad (1)$$

**Our Contributions.** In this paper we construct an algorithm that produces an estimate $\hat{W}$ from a single instance $G_n$ of a $W$-random graph with target density $\rho_n$ (or simply $\rho$, when $n$ is clear from the context). We aim for several properties:

1. $\hat{W}$ is differentially private;
2. $\hat{W}$ is consistent, in the sense that $\delta_2(W, \hat{W}) \to 0$ in probability as $n \to \infty$;
3. $\hat{W}$ has a compact representation (in our case, as a matrix with $o(n)$ entries);
4. The procedure works for sparse graphs, that is, when the density $\rho$ is small;
5. On input $G_n$, $\hat{W}$ can be calculated efficiently.

Here we give an estimation procedure that obeys the first four properties, leaving the question of polynomial-time algorithms for future work. Given an input graph $G_n$, a privacy-parameter $\epsilon$ and a target number $k$ of blocks, our algorithm $\mathcal{A}$ produces a $k$-block graphon $\hat{W} = \mathcal{A}(G_n)$ such that

- $\mathcal{A}$ is $\epsilon$-differentially node private. The privacy guarantee holds for all inputs, independent of modeling assumptions.
- If (1) $W$ is an arbitrary graphon, normalized so $\int W = 1$, (2) the expected average degree $(n-1)\rho$ grows at least as fast as $\log n$, and (3) $k$ goes to infinity sufficiently slowly with $n$, then, when $G_n$ is $\rho W$-random, the estimate $\hat{W}$ for $W$ is *consistent* (that is, $\delta_2(\hat{W}, W) \to 0$, both in probability and almost surely).
- We give a nonprivate variant of $\mathcal{A}$ that converges assuming only $\omega(1)$ average degree.

Combined with the general theory of convergent graphs sequences, these results in particular give a node-private procedure for estimating the edge density of all cuts in a $\rho W$-random graph, see Section 2.2 below.

The main idea of our algorithm is to use the exponential mechanism of [25] to select a block model which approximately minimizes the $\ell_2$ distance to the observed adjacency matrix of $G$, under the best possible assignment of nodes to blocks (this explicit search over assignments makes the algorithm take exponential time). In order to get an algorithm that is accurate on sparse graphs, we need several nontrivial extensions of current techniques. To achieve privacy, we use a new variation of the Lipschitz extension technique of [21, 14] to reduce the sensitivity of the $\delta_2$ distance. While those works used Lipschitz extensions for noise addition, we use of Lipschitz extensions inside the "exponential mechanism" [25] (to control the sensitivity of the score functions). To bound our algorithm's error, we provide a new analysis of the $\ell_2$-minimization algorithm; we show that approximate minimizers are not too far from the actual minimizer (a "stability" property). Both aspects of our work are enabled by restricting the $\ell_2^2$-minimization to a set of block models whose density (in fact, $L_\infty$ norm) is not much larger than that of the underlying graph. The algorithm is presented in Section 3.

Our most general result proves consistency for arbitrary graphons $W$ but does not provides a concrete rate of convergence. However, we provide explicit rates under various assumptions on $W$. Specifically, we relate the error of our estimator to two natural error terms involving the graphon $W$: the error $\epsilon_k^{(O)}(W)$ of the best $k$-block approximation to $W$ in the $L_2$ norm (see (4) below) and an error term $\epsilon_n(W)$ measuring the $L_2$-distance between the graphon $W$ and the matrix of probabilities $H_n(W)$ generating the graph $G_n$ (see (5) below.) In terms of these error terms, Theorem 1 shows

$$\delta_2\left(W, \hat{W}\right) \le \epsilon_k^{(O)}(W) + 2\epsilon_n(W) + O_P\left(\sqrt[4]{\frac{\log k}{\rho n}} + \sqrt{\frac{k^2 \log n}{n\epsilon}} + \frac{1}{\rho\epsilon n}\right). \qquad (2)$$

provided the average degree $\rho n$ grows at least like $\log n$. Along the way, we provide a novel analysis of a straightforward, nonprivate least-squares estimator that does not require an assumption on the average degree, and leads to an error bound with a better dependence on $k$:

$$\delta_2\left(W, \hat{W}_{\text{nonprivate}}\right) \le \epsilon_k^{(O)}(W) + 2\epsilon_n(W) + O_P\left(\sqrt[4]{\frac{\log k}{\rho n}} + \frac{k^2}{\rho n^2}\right). \qquad (3)$$

It follows from the theory of graph convergence that for all graphons $W$, we have $\epsilon_k^{(O)}(W) \to 0$ as $k \to \infty$ and $\epsilon_n(W) \to 0$ almost surely as $n \to \infty$. By selecting $k$ appropriately, the nonprivate algorithm converges for any bounded graphon as long as $\rho n \to \infty$ with $n$; the private algorithm converges whenever $\rho n \ge 6 \log n$ (e.g., for constant $\epsilon$). As proven in the full version, we also have $\epsilon_n(W) = O_P(\epsilon_k^{(O)}(W) + \sqrt[4]{k/n})$, though this upper bound is loose in many cases.

As a specific instantiation of these bounds, let us consider the case that $W$ is exactly described by a $k$-block model, in which case $\epsilon_k^{(O)}(W) = 0$ and $\epsilon_n(W) = O_P(\sqrt[4]{k/n})$ (see full version for a proof). For $k \leq (n/\log^2 n)^{1/3}$, $\rho \geq \log(k)/k$ and constant $\epsilon$, our private estimator has an asymptotic error that is dominated by the (unavoidable) error of $\epsilon_n(W) = \sqrt[4]{k/n}$, showing that we do not lose anything due to privacy in this special case. Another special case is when $W$ is $\alpha$-Hölder continuous, in which case $\epsilon_k^{(O)}(W) = O(k^{-\alpha})$ and $\epsilon_n(W) = O_P(n^{-\alpha/2})$; see Remark 2 below.

**Comparison to Previous Nonprivate Bounds.** We provide the first consistency bounds for estimation of a nonparametric graph model subject to node differential privacy. Along the way, for sparse graphs, we provide more general consistency results than were previously known, regardless of privacy. In particular, to the best of our knowledge, *no prior results give a consistent estimator for $W$ that works for sparse graphs without any additional assumptions besides boundedness.*

When compared to results for nonprivate algorithms applied to graphons obeying additional assumptions, our bounds are often incomparable, and in other cases match the existing bounds.

We start by considering graphons which are themselves step functions with a known number of steps $k$. In the dense case, the nonprivate algorithms of [18] and [13], as well as our nonprivate algorithm, give an asymptotic error that is dominated by the term $\epsilon_n(W) = O(\sqrt[4]{k/n})$, which is of the same order as our private estimator as long as $k = \tilde{o}(n^{1/3})$. [28] provided the first convergence results for estimating graphons in the sparse regime. Assuming that $W$ is bounded above and below (so it takes values in a range $[\lambda_1, \lambda_2]$ where $\lambda_1 > 0$), they analyze an inefficient algorithm (the MLE). The bounds of [28] are incomparable to ours, though for the case of $k$-block graphons, both their bounds and our nonprivate bound are dominated by the term $\sqrt[4]{k/n}$ when $\rho > (\log k)/k$ and $k \leq \rho n$. A different sequence of works shows how to consistently estimate the underlying block model with a *fixed* number of blocks $k$ in polynomial time for very sparse graphs (as for our non-private algorithm, the only thing which is needed is that $n\rho \to \infty$) [3, 1, 2]; we are not aware of concrete bounds on the convergence rate.

For the case of *dense $\alpha$-Hölder-continuous graphons*, the results of [18] give an error which is dominated by the term $\epsilon_n(W) = O_P(n^{-\alpha/2})$. For $\alpha < 1/2$, our nonprivate bound matches this bound, while for $\alpha > 1/2$ it is worse. [28] considers the sparse case. The rate of their estimator is incomparable to that of ours; further, their analysis requires a lower bound on the edge probabilities, while ours does not. Very recently, after our paper was submitted, both the bounds of [28] as well as our non-private bound (3) were substantially improved [22], leading to an error bound where the 4th root in (3) is replaced by a square root (at the cost of an extra constant multiplying the oracle error.)

See the full version for a more detailed discussion of the previous literature.

## 2 Preliminaries

### 2.1 Notation

For a graph $G$ on $[n] = \{1, \ldots, n\}$, we use $E(G)$ and $A(G)$ to denote the edge set and the adjacency matrix of $G$, respectively. The edge density $\rho(G)$ is defined as the number of edges divided by $\binom{n}{2}$. Finally the degree $d_i$ of a vertex $i$ in $G$ is the number of edges containing $i$. We use the same notation for a weighted graph with nonnegative edge weights $\beta_{ij}$, where now $\rho(G) = \frac{2}{n(n-1)} \sum_{i<j} \beta_{ij}$, and $d_i = \sum_{j \neq i} \beta_{ij}$. We use $\mathbb{G}_n$ to denote the set of weighted graphs on $n$ vertices with weights in $[0,1]$, and $\mathbb{G}_{n,d}$ to denote the set of all graphs in $\mathbb{G}_n$ that have maximal degree at most $d$.

**From Matrices to Graphons.** We define a graphon to be a bounded, measurable function $W : [0,1]^2 \to \mathbb{R}_+$ such that $W(x,y) = W(y,x)$ for all $x, y \in [0,1]$. It will be convenient to embed the set of a symmetric $n \times n$ matrix with nonnegative entries into graphons as follows: let $\mathcal{P}_n = (I_1, \ldots I_n)$ be the partition of $[0,1]$ into adjacent intervals of lengths $1/n$. Define $W[A]$ to be the step function which equals $A_{ij}$ on $I_i \times I_j$. If $A$ is the adjacency matrix of an unweighted graph $G$, we use $W[G]$ for $W[A]$.

**Distances.** For $p \in [1, \infty)$ we define the $L_p$ norm of an $n \times n$ matrix $A$ and a (Borel)-measurable function $W : [0,1]^2 \to \mathbb{R}$ by $\|A\|_p = \left( \frac{1}{n^2} \sum_{i,j} |A_{ij}|^p \right)^{1/p}$, and $\|f\|_p = \left( \int |f(x,y)|^p dxdy \right)^{1/p}$,

respectively. Associated with the $L_2$-norm is a scalar product, defined as $\langle A, B \rangle = \frac{1}{n^2} \sum_{i,j} A_{ij} B_{ij}$ for two $n \times n$ matrices $A$ and $B$, and $\langle U, W \rangle = \int U(x,y) W(x,y) dx dy$ for two square integrable functions $U, W : [0,1]^2 \to \mathbb{R}$. Note that with this notation, the edge density and the $L_1$ norm are related by $\|G\|_1 = \frac{n-1}{n} \rho(G)$.

Recalling (1), we define the $\delta_2$ distance between two matrices $A, B$, or between a matrix $A$ and a graphon $W$ by $\delta_2(A, B) = \delta_2(W[A], W[B])$ and $\delta_2(A, W) = \delta_2(W[A], W)$. In addition, we will also use the in general larger distances $\hat{\delta}_2(A, B)$ and $\hat{\delta}_2(A, W)$, defined by taking a minimum over matrices $A'$ which are obtained from $A$ by a relabelling of the indices: $\hat{\delta}_2(A, B) = \min_{A'} \|A' - B\|_2$ and $\hat{\delta}_2(A, W) = \min_{A'} \|W[A'] - W\|_2$.

## 2.2  $W$-random graphs, graph convergence and multi-way cuts

**W-random graphs and stochastic block models.** Given a graphon $W$ we define a random $n \times n$ matrix $H_n = H_n(W)$ by choosing $n$ "positions" $x_1, \ldots, x_n$ i.i.d. uniformly at random from $[0,1]$ and then setting $(H_n)_{ij} = W(x_i, x_j)$. If $\|W\|_\infty \le 1$, then $H_n(W)$ has entries in $[0,1]$, and we can form a random graph $G_n = G_n(W)$ on $n$-vertices by choosing an edge between two vertices $i < j$ with probability $(H_n)_{ij}$, independently for all $i < j$. Following [23] we call $G_n(W)$ a $W$-random graph and $H_n(W)$ a $W$-weighted random graph. We incorporate a target density $\rho_n$ (or simply $\rho$, when $n$ is clear from the context) by normalizing $W$ so that $\int W = 1$ and taking $G$ to be a sample from $G_n(\rho W)$. In other words, we set $Q = H_n(\rho W) = \rho H_n(W)$ and then connect $i$ to $j$ with probability $Q_{ij}$, independently for all $i < j$.

Stochastic block models are specific examples of $W$-random graph in which $W$ is constant on sets of the form $I_i \times I_j$, where $(I_1, \ldots, I_k)$ is a partition of $[0,1]$ into intervals of possibly different lengths.

On the other hand, an arbitrary graphon $W$ can be well approximated by a block model. Indeed, let

$$\epsilon_k^{(O)}(W) = \min_B \|W - W[B]\|_2 \tag{4}$$

where the minimum runs over all $k \times k$ matrices $B$. By a straightforward argument (see, e.g., [11]) $\epsilon_k^{(O)}(W) = \|W - W_{\mathcal{P}_k}\|_2 \to 0$ as $k \to \infty$. We will take this approximation as a benchmark for our approach, and consider it the error an "oracle" could obtain (hence the superscript $O$).

Another key term in our algorithm's error guarantee is the distance between $H_n(W)$ and $W$,

$$\epsilon_n(W) = \hat{\delta}_2(H_n(W), W). \tag{5}$$

It goes to zero as $n \to \infty$ by the following lemma, which follows easily from the results of [11].

**Lemma 1.** *Let $W$ be a graphon with $\|W\|_\infty < \infty$. With probability one, $\|H_n(W)\|_1 \to \|W\|_1$ and $\epsilon_n(W) \to 0$.*

**Convergence.** Given a sequence of $W$-random graphs with target densities $\rho_n$, one might wonder whether the graphs $G_n = G_n(\rho_n W)$ converge to $W$ in a suitable metric. The answer is yes, and involves the so-called cut-metric $\delta_\square$ first introduced in [9]. Its definition is identical to the definition (1) of the norm $\delta_2$, except that instead of the $L_2$-norm $\|\cdots\|_2$, it involves the Frieze-Kannan cut-norm $\|W\|_\square$ defined as the sup of $\left| \int_{S \times T} W \right|$ over all measurable sets $S, T \subset [0,1]$. In the metric $\delta_\square$, the $W$-random graphs $G_n = G_n(\rho W)$ then converge to $W$ in the sense that $\delta_\square \left( \frac{1}{\rho(G_n)} W[G_n], W \right) \to 0$, see [11] for the proof.

**Estimation of Multi-Way Cuts.** Using the results of [12], the convergence of $G_n$ in the cut-metric $\delta_\square$ implies many interesting results for estimating various quantities defined on the graph $G_n$. Indeed, a consistent approximation $\hat{W}$ to $W$ in the metric $\delta_2$ is clearly consistent in the weaker metric $\delta_\square$. This distance, in turn, controls various quantities of interest to computer scientists, e.g., the size of all multi-way cuts, implying that a consistent estimator for $W$ also gives consistent estimators for all multi-way cuts. See the full version for details.

### 2.3 Differential Privacy for Graphs

The goal of this paper is the development of a differentially private algorithm for graphon estimation. The privacy guarantees are formulated for worst-case inputs — we do not assume that $G$ is generated from a graphon when analyzing privacy. This ensures that the guarantee remains meaningful no matter what an analyst knows ahead of time about $G$.

In this paper, we consider node privacy. We call two graphs $G$ and $G'$ *node neighbors* if one can be obtained from the other by removing one node and its adjacent edges.

**Definition 1** ($\epsilon$-node-privacy). *A randomized algorithm $\mathcal{A}$ is $\epsilon$-node-private if for all events $S$ in the output space of $\mathcal{A}$, and node neighbors $G, G'$,*

$$\Pr[\mathcal{A}(G) \in S] \leq \exp(\epsilon) \times \Pr[\mathcal{A}(G') \in S].$$

We also need the notion of the *node-sensitivity* of a function $f : \mathbb{G}_n \to \mathbb{R}$, defined as maximum $\max_{G, G'} |f(G) - f(G')|$, where the maximum goes over node-neighbors. The node sensitivity is the Lipshitz constant of $f$ viewed as a map between appropriate metrics.

## 3 Differentially Private Graphon Estimation

### 3.1 Least-squares Estimation

Given a graph as input generated by an unknown graphon $W$, our goal is to recover a block-model approximation to $W$. The basic nonprivate algorithm we emulate is least squares estimation, which outputs the $k \times k$ matrix $B$ which is closest to the input adjacency matrix $A$ in the distance

$$\hat{\delta}_2(B, A) = \min_{\pi} \|B_{\pi} - A\|_2,$$

where the minimum runs over all equipartitions $\pi$ of $[n]$ into $k$ classes, i.e., over all maps $\pi : [n] \to [k]$ such that all classes have size as close to $n/k$ as possible, i.e., such that $||\pi^{-1}(i)| - n/k| < 1$ for all $i$, and $B_{\pi}$ is the $n \times n$ block-matrix with entries $(B_{\pi})_{xy} = B_{\pi(x)\pi(y)}$. If $A$ is the adjacency matrix of a graph $G$, we write $\hat{\delta}_2(B, G)$ instead of $\hat{\delta}_2(B, A)$. In the above notation, the basic algorithm we would want to emulate is then the algorithm which outputs the least square fit $\hat{B} = \operatorname{argmin}_B \hat{\delta}_2(B, G)$, where the argmin runs over all symmetric $k \times k$ matrices $B$.

### 3.2 Towards a Private Algorithm

Our algorithm uses a carefully chosen instantiation of the *exponential mechanism* of McSherry and Talwar [25]. The most direct application of their framework would be to output a random $k \times k$ matrix $\hat{B}$ according to the probability distribution

$$\Pr(\hat{B} = B) \propto \exp\left(-C\hat{\delta}_2^2(B, A)\right),$$

for some $C > 0$. The resulting algorithm is $\epsilon$-differentially private if we set $C$ to be $\epsilon$ over twice the node-sensitivity of the "score function", here $\delta_2^2(B, \cdot)$. But this value of $C$ turns out to be too small to produce an output that is a good approximation to the least square estimator. Indeed, for a given matrix $B$ and equipartition $\pi$, the node-sensitivity of $\|G - B_{\pi}\|_2^2$ can be as large as $\frac{1}{n}$, leading to a value of $C$ which is too small to produce useful results for sparse graphs.

To address this, we first note that we can work with an equivalent score that is much less sensitive. Given $B$ and $\pi$, we subtract off the squared norm of $G$ to obtain the following:

$$
\begin{align}
score(B, \pi; G) &= \|G\|_2^2 - \|G - B_{\pi}\|_2^2 = 2\langle G, B_{\pi}\rangle - \|B_{\pi}\|^2, \text{ and} \tag{6} \\
score(B; G) &= \max_{\pi} score(B, \pi; G), \tag{7}
\end{align}
$$

where the max ranges over equipartitions $\pi : [n] \to [k]$. For a fixed input graph $G$, maximizing the score is the same as minimizing the distance, i.e. $\operatorname{argmin}_B \hat{\delta}_2(B, G) = \operatorname{argmax}_B score(B; G)$.

The sensitivity of the new score is then bounded by $\frac{2}{n^2} \cdot \|B\|_\infty$ times the maximum degree in $G$ (since $G$ only affects the score via the inner product $\langle G, B_\pi \rangle$). But this is still problematic since, a priori, we have no control over either the size of $\|B\|_\infty$ or the maximal degree of $G$.

To keep the sensitivity low, we make two modifications: first, we only optimize over matrices $B$ whose entries bounded by (roughly) $\rho_n$ (since a good estimator will have entries which are not much larger than $\|\rho_n W\|_\infty$, which is of order $\rho_n$); second, we restrict the score to be accurate only on graphs whose maximum degree is at most a constant times the average degree, since this is what one expects for graphs generated from a bounded graphon. While the first restriction can be directly enforced by the algorithm, the second is more delicate, since we need to provide privacy for *all* inputs, including graphs with very large maximum degree. We employ an idea from [6, 21]: we first consider the restriction of $score(B, \pi; \cdot)$ to $\mathbb{G}_{n,d_n}$ where $d_n$ will be chosen to be of the order of the average degree of $G$, and then extend it back to all graphs while keeping the sensitivity low.

### 3.3 Private Estimation Algorithm

Our final algorithm takes as input the privacy parameter $\epsilon$, the graph $G$, a number $k$ of blocks, and a constant $\lambda \geq 1$ that will have to be chosen large enough to guarantee consistency of the algorithm.

---

**Algorithm 1:** Private Estimation Algorithm

---

**Input**: $\epsilon > 0$, $\lambda \geq 1$, an integer $k$ and graph $G$ on $n$ vertices.

**Output**: $k \times k$ block graphon (represented as a $k \times k$ matrix $\hat{B}$) estimating $\rho W$

Compute an $(\epsilon/2)$-node-private density approximation $\hat{\rho} = \rho(G) + \text{Lap}(4/n\epsilon)$ ;

$d = \lambda \hat{\rho} n$ (the target maximum degree) ;

$\mu = \lambda \hat{\rho}$ (the target $L_\infty$ norm for $\hat{B}$) ;

For each $B$ and $\pi$, let $\widehat{score}(B, \pi; \cdot)$ denote a nondecreasing Lipschitz extension (from [21]) of $score(B, \pi; \cdot)$ from $\mathbb{G}_{n,d}$ to $\mathbb{G}_n$ such that for all matrices $A$, $\widehat{score}(B, \pi; A) \leq score(B, \pi; A)$, and define

$$\widehat{score}(B; A) = \max_\pi \widehat{score}(B, \pi; A)$$

**return** $\hat{B}$, sampled from the distribution

$$\Pr(\hat{B} = B) \propto \exp\left( \frac{\epsilon}{4\Delta} \widehat{score}(B; A) \right),$$

where $\Delta = \frac{4d\mu}{n^2} = \frac{4\lambda^2 \hat{\rho}^2}{n}$ and $B$ ranges over matrices in

$$\mathcal{B}_\mu = \{ B \in [0, \mu]^{k \times k} : \text{all entries } B_{i,j} \text{ are multiples of } \tfrac{1}{n} \};$$

---

Our main results about the private algorithm are the following lemma and theorem.

**Lemma 2.** *Algorithm 1 is $\epsilon$-node private.*

**Theorem 1** (Performance of the Private Algorithm). *Let $W : [0,1]^2 \to [0, \Lambda]$ be a normalized graphon, let $0 < \rho\Lambda \leq 1$, let $G = G_n(\rho W)$, $\lambda \geq 1$, and $k$ be an integer. Assume that $\rho n \geq 6 \log n$ and $8\Lambda \leq \lambda \leq \sqrt{n}$, $2 \leq k \leq \min\{n\sqrt{\frac{\rho}{2}}, e^{\frac{\rho n}{2}}\}$. Then the Algorithm 1 outputs an approximation $(\hat{\rho}, \hat{B})$ such that*

$$\delta_2\left(W, \frac{1}{\hat{\rho}} W[\hat{B}]\right) \leq \epsilon_k^{(O)}(W) + 2\epsilon_n(W) + O_P\left( \sqrt[4]{\frac{\lambda^2 \log k}{\rho n}} + \lambda \sqrt{\frac{k^2 \log n}{n\epsilon}} + \frac{\lambda}{n\rho\epsilon} \right).$$

**Remark 1.** *While Theorem 1 is stated in term of bounds which hold in probability, our proofs yield statements which hold almost surely as $n \to \infty$.*

**Remark 2.** *Under additional assumptions on the graphon $W$, we obtain tighter bounds. For example, if we assume that $W$ is Hölder continuous, i.e, there exist constants $\alpha \in (0, 1]$ and $C < \infty$ such that $|W(x, y) - W(x', y')| \leq C\delta^\alpha$ whenever $|x - x'| + |y - y'| \leq \delta$, then we have that $\epsilon_k^{(O)}(W) = O(k^{-\alpha})$ and $\epsilon_n(W) = O_P(n^{-\alpha/2})$.*

**Remark 3.** *When considering the "best" block model approximation to $W$, one might want to consider block models with unequal block sizes; in a similar way, one might want to construct a private algorithm that outputs a block model with unequal size blocks, and produces a bound in terms of this best block model approximation instead of $\epsilon_k^{(O)}(W)$. This can be proved with our methods, with the minimal block size taking the role of $1/k$ in all our statements.*

### 3.4 Non-Private Estimation Algorithm

We also analyze a simple, non-private algorithm, which outputs the argmin of $\hat{\delta}_2(\cdot, A)$ over all $k \times k$ matrices whose entries are bounded by $\lambda\rho(G)$. (Independently of our work, this non-private algorithm was also proposed and analysed in [22].) Our bound (3) refers to this restricted least square algorithm, and does not require any assumptions on the average degree. As in (2), we suppress the dependence of the error on $\lambda$. To include it, one has to multiply the $O_P$ term in (3) by $\sqrt{\lambda}$.

## 4 Analysis of the Private and Non-Private Algorithm

At a high level, our proof of Theorem 1 (as well as our new bounds on non-private estimation) follow from the fact that for all $B$ and $\pi$, the expected score $\mathbb{E}[Score(B, \pi; G)]$ is equal to the score $Score(B, \pi; Q)$, combined with a concentration argument. As a consequence, the maximizer $\hat{B}$ of $Score(B; G)$ will approximately minimize the $L_2$-distance $\hat{\delta}_2(B, Q)$, which in turn will approximately minimize $\|\frac{1}{\rho}W[B] - W\|_2$, thus relating the $L_2$-error of our estimator $\hat{B}$ to the "oracle error" $\epsilon_k^{(O)}(W)$ defined in (4).

Our main concentration statement is captured in the following proposition. To state it, we define, for every symmetric $n \times n$ matrix $Q$ with vanishing diagonal, $Bern_0(Q)$ to be the distribution over symmetric matrices $A$ with zero diagonal such that the entries $\{A_{ij} : i < j\}$ are independent Bernouilli random variables with $\mathbb{E}A_{ij} = Q_{ij}$.

**Proposition 1.** *Let $\mu > 0$, $Q \in [0, 1]^{n \times n}$ be a symmetric matrix with vanishing diagonal, and $A \sim Bern_0(Q)$. If $2 \le k \le \min\{n\sqrt{\rho(Q)}, e^{\rho(Q)n}\}$ and $\hat{B} \in \mathcal{B}_\mu$ is such that*

$$Score(\hat{B}; A) \ge \max_{B \in \mathcal{B}_\mu} Score(B; A) - \nu^2$$

*for some $\nu > 0$, then with probability at least $1 - 2e^{-n}$,*

$$\hat{\delta}_2(\hat{B}, Q) \le \min_{B \in \mathcal{B}_\mu} \hat{\delta}_2(B, Q) + \nu + O\left(\sqrt[4]{\mu^2 \rho(Q)\left(\frac{k^2}{n^2} + \frac{\log k}{n}\right)}\right). \tag{8}$$

Morally, the proposition contains almost all that is needed to establish the bound (3) proving consistency of the non-private algorithm (which, in fact, only involves the case $\nu = 0$), even though there are several additional steps needed to complete the proof.

The proposition also contains an extra ingredient which is a crucial input for the analysis of the private algorithm: it states that if instead of an optimal, least square estimator, we output an estimator whose score is only approximately maximal, then the excess error introduced by the approximation is small. To apply the proposition, we then establish a lemma which gives us a lower bound on the score of the output $\hat{B}$ in terms of the maximal score and an excess error $\nu$.

There are several steps needed to execute this strategy, the most important ones involving a rigorous control of the error introduced by the Lipschitz extension inside the exponential algorithm. We defer the details to the full version.

**Acknowledgments.** A.S. was supported by NSF award IIS-1447700 and a Google Faculty Award. Part of this work was done while visiting Boston University's Hariri Institute for Computation and Harvard University's Center for Research on Computation and Society.

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
