[Reviews · NeurIPS 2015]

Submitted by Assigned_Reviewer_1

This paper considers the node private graphon estimation problem. For general graphon, the authors prove the consistency result. For graphon with additional structure, explicit rates of convergence are provided. The authors also compare their obtained rates of convergence with those obtained by nonprivate estimators.

Overall, I think this paper is very clearly written and the contribution is quite solid.
Summary: This is a solid theoretical paper on differentially private graphon estimation. I think the authors made an important contribution.

Submitted by Assigned_Reviewer_2

The paper deals with

learning graphons, limits of large graphs, represented as symmetric functions on a square, from an input graph. In particular, the authors deal with differentially private learning so that small variations of the input cannot be reconstructed from the obtained model. The paper appears to be theoretically solid and is clearly written.

I am not sufficiently familiar with the literature to judge the technical contributions in terms of the techniques and rate improvements in the not-private regime.

I think this is a reasonable paper and can appear in the conference.

My main concern is that estimating graphon in a

differentially private manner seems to be a somewhat esoteric problem. I would like the authors to convince me that such estimation can be of interest to more than a few specialists by providing a setting where this problem could appear.

In addition, the main contribution of the paper is purely theoretical, with the proposed algorithm

required to enumerate all partitions. The authors say that leave the question of efficiency to future work. Still, perhaps an argument can be made that more practical algorithms are possible or, alternatively, that this is an essential limitation?

Summary: The paper is clearly written and appears to be theoretically solid. My main concern is that the problem is somewhat obscure and the contribution is purely theoretical (the main algorithm needs to run over all partitions).

Submitted by Assigned_Reviewer_3

This paper studies the problem of differential privately estimation of a graphon. The paper achieves many desirable properties, including node-differential privacy (much better than edge-differential privacy), consistent estimate even for sparse graphs, and fairly weak assumptions on the graphon.

The main weakness of the paper is that the result is not algorithmic. This allows the paper to get around two issues: 1. how to efficiently estimate a graphon in the sparse regime and 2. when there is no running time requirement the way to do differential privacy is "almost known": just use the exponential mechanism, for many previous works in differential privacy it is trivial to get an algorithm with huge running time and the main challenge was to make it efficient. However in this paper it feels like the problem is already fairly complicated that even analyzing the exponential mechanism is nontrivial (and indeed the paper needs to use a carefully designed score function instead of the naive one), and there are some interesting ideas and observations that goes into the proof.

Overall I think this is still interesting even though the lack of algorithm is a major concern. The authors should emphasize that in the abstract, as when you are claiming "our bounds improve on nonprivate results" part of the reason is there is no efficient algorithm.
Summary: The paper gives strong results for the problem except there is no corresponding algorithm. The result is still interesting.

Submitted by Assigned_Reviewer_4

The main theoretical contributions are: (i) in the finite graph setting, the authors provide an analysis of the L2 estimation quality of symmetric probability matrices using an ideal stochastic block model estimator (ii) the analysis is extended to a consistency analysis of the graphon parameter in a sparse graph generative model (combined with a sample-size dependent sparsity parameter), again using the L2 loss (iii) for both cases the analysis is extended to propose a mechanism for differentially private graph estimation. Each of these results is interesting and important in its own right, and advance the state of the art under what seem like reasonable conditions. Overall, the manuscript well written and conveys the main ideas, though it is clearly compressed from a full version.

The "longer version" mentioned in the manuscript they refers to the supplement - which is an extended version of the submitted paper. While I like that the supplement is self contained, this is non-standard for a conference paper. If this paper is accepted, I suggest that the authors modify the appendix to the standard format. - Some standard elements e.g. a conclusion, statement of future work, and some (possibly preliminary) experimental evaluation are not included, I imagine due to limited space, or differences in writing styles/expectations across communities?

- The metric on line 219 seems to me comparing a matrix A with a graphon function W using the L2 error. This does not make sense to me. I suspect it is simply a mistake in the definition. Should the definition be min_{A'} || W[A']-W ||?

- It will be useful for the authors to specify explicit conditions (if any) under which the function line: 308 is normalizable as a density.

Minor comments:

- The definitions of L_p norm in lines 209, 210 are quite non-standardize. the authors use the p^th power of the standard definition. I suggest the authors clarify this choice in the manuscript.

- Line 234: Authors should note which theorem in [7] they refer to.

- Could the authors expand on an explicit algorithm for sampling from the density (line: 308) e.g. an importance sampling or Metropolis-Hastings based approach?

- Line 284: "small" g' should be "capital" G.
Summary: The authors propose an estimator for graph data which satisfies node-differentially private guarantees with respect to a modified L2 error. The manuscript is clearly written with interesting and useful results which will be of further theoretical and practical interest.

Author Feedback
Author rebuttal: We thank the reviewers for their thoughtful comments.

General comments: The reviewers' main concerns had to do with the importance of the problem and the efficiency of the (private and nonprivate) algorithms.

A) Importance of the problem. Several reviewers mentioned that the problem of private graphon estimation is esoteric. We disagree.

Graphons are an extremely general model of exchangeable distributions on graphs and are now the subject of intense study in the statistics community. Any graph model in which edge probabilities are functions of an underlying "type" for each vertex, and the vertex types are i.i.d., can be expressed as a graphon (that is, any "type space" can be mapped to the uniform distribution on [0,1]). Essentially *any* graph distribution which is exchangeable (i.e. symmetric under vertex permutations) and consistent under subsampling (i.e. the model is closed under taking subgraphs) is a graphon. Graphons play the same role in distributions on graphs as i.i.d. sequences play in distributions on sequences. In fact, the celebrated Szemeredi Regularity Lemma says that any graph is close (in the cut norm) to a block graphon, uniformly in the size of the graph, provided the graphon has enough blocks.

Block graphons are the same as stochastic block models, a powerful and very widely-studied class of graph distributions. We provide, in particular, new results on the consistency of estimating stochastic block models with a large number of bocks.

Mixed membership models can also be written as W-random graphs; indeed, in the simplest incarnation, these can described as follows: for each vertex, choose an element p from the simplex \Delta_k according to a Dirichlet distribution, and then define W(p,p')=\sum_{i,j} p_i p'_j. Using a measure preserving transformation from the simplex into the unit interval, one obtains an equivalent graphon over [0,1]. [The above example also points to the importance of not assuming continuity of graphons, at least when one insists to describe them as functions of variables in [0,1], since many interesting examples of graphons are defined over latent position spaces which are not [0,1], and have to be transformed to this space via transformations which don't preserve continuity.]

Privacy in the analysis of graph data is a widely recognized and difficult problem. We did not provide extensive background and citations due the space constraints (especially the one-page limit on citations). We will try to address that in future versions. However, we note that there are many techniques for re-identifying individuals from supposedly anonymous network data (for example, the works of Backstrom, Dwork and Kleinberg, 2007 and Narayanan and Shmatikov 2009). The design of accurate methods for analyzing network data with rigorous privacy guarantees is challenging, with dozens of papers on the topic appearing in the last decade in top venues in related fields such as databases (SIGMOD), theory (STOC, TCC, CRYPTO) and data mining (KDD), among others.

On a related note, it *is* true that the NIPS community has not devoted a lot of attention to privacy issues (in social networks or elsewhere), although there was a plenary and a widely attended workshop on differential privacy methods during the 2014 NIPS. Part of our goal in submitting to NIPS is to raise awareness of the issue in the ML community-a community that is critical for developing useful, scalable techniques for analyzing sensitive "big" data.

B) Efficiency. We agree that the design of efficient algorithms is critical, but we feel that feasibility results are important steps.

Designing (private or nonprivate) algorithms for this problem is nontrivial even without efficiency (even the use of the exponential mechanism here is nontrivial). The algorithms studied in the literature (such as in works we cite, and the recent work of Abbe and Standon) are neither efficient nor simple to analyze.

Results on the feasibility of estimation are widely studied in the statistics community, are important to understanding what one can hope to eventually achieve with efficient algorithms. In particular, the results in this paper are the strongest existing results on feasibility

--

The reviewers also provided a number of specific comments on presentation and typos. We appreciate the detailed feedback and look forward to incorporating it in future versions of the paper.